# Psychosocial well-being and mental health of low- and middle-income countries' internally displaced persons and refugees during COVID-19: a systematic literature review

Oluwakemi Amodu[1] ⬤, Craig R. Janes[2] and Karen Therese L. Pangan[1] ⬤

[1]Faculty of Nursing, University of Alberta and [2]Faculty of Health of the University of Waterloo

## Overview Review

COVID-19; global mental health; mental well-being; refugee; displacement

**Corresponding author:**
Karen Therese L. Pangan;
Email: karenthe@ualberta.ca

## Abstract

**Background:** The COVID-19 pandemic brought to light the need to address the psychosocial and mental health needs of refugees and internally displaced persons in low- and middle-income countries. COVID-19 prevention measures slowed essential services and healthcare, creating unique challenges for refugees and IDPs, including economic insecurity and societal instability. All of these factors may contribute to the reported declines in their psychosocial well-being. **Methods:** To effectively define the problems of low-and middle-income countries (LMICs) in addressing the needs of these populations, we conducted a systematic literature review of literature on the mental health and psychosocial well-being of refugees and displaced persons who have migrated between LMICs in the context of COVID-19. **Findings:** Our findings indicate that mental health interventions, such as digital healthcare and community-focused solutions, have the potential to address the problems faced by refugees and IDPs. Nevertheless, these community-based support networks are overextended, continuously developing to meet the needs of these vulnerable populations while considering the limited digital literacy of the subject population, internet accessibility, and overall limits in reach. We found that the efficacy of interventions varied according to the distinctive needs and challenges of various refugee and IDP populations. **Implications:** The findings indicate a need for an intersectional policy approach to address the complex network of factors influencing mental health outcomes, including gender, housing, employment status, and social inequalities. Global agencies, policymakers, and local governments must prioritize the development of comprehensive mental health support systems, assuring refugees and IDPs have sustainable and equitable access.

## Impact statement

In low- and middle-income countries, the COVID-19 pandemic has significantly impacted the psychological and mental health of refugees and internally displaced persons (IDPs). These vulnerable groups face increased risks to their mental health due to the pandemic aggravating preexisting issues, such as restricted access to resources and economic uncertainty. This study has brought attention to the demand for innovative, situation-specific responses that consider the difficulties and cultural backgrounds of refugees and IDPs.

Digital solutions and community-focused therapies have shown some potential to tackle the difficulties faced by these groups, though further research is required to determine whether such interventions are effective and sustainable. Moving forward, global groups, decision-makers, and local authorities must prioritize the locally important social and economic factors that affect the mental health and psychosocial welfare of refugees and IDPs during and after the pandemic. This involves utilizing digital technologies and tailoring treatments to specific country contexts. Finally, to address the mental health needs of refugees and IDPs in low- and middle-income countries, both during a crisis and in anticipation of future challenges, a holistic and long-term approach to improving living conditions, income security, and access to essential services remains crucial.

## Introduction

The World Health Organization (WHO) acknowledges that the COVID-19 pandemic, which began in March 2020, has taken a huge toll on the mental health and well-being of populations worldwide. The epidemic of mental health problems at its intersection with COVID-19 presents a monumental challenge to the public health systems across high-income and low-income countries (WHO, 2020a). As the virus continued to spread globally, government-imposed

containment strategies challenged the global economy and the health and essential service infrastructure. In response to the rise of COVID-19, health and social care workers were redeployed to frontline service centers to care for COVID-19 patients and health facilities were converted to COVID-19 quarantine and treatment points. Unfortunately, the swift induction of these measures led to a de-prioritization of nonurgent health service delivery, including mental health and substance use services.

### The effects of COVID-19 on support for refugees and displaced persons

Changes to the economy and government infrastructure caused by the pandemic have highlighted the unequal impact that a crisis can have on the most vulnerable populations, including the elderly, women and children, minorities, the sick, and the displaced. This impact is further exacerbated in displaced populations who have moved from their place of origin in hopes of finding better livelihoods or fleeing violence and oppression, including refugees and internally displaced persons. An examination of the challenges faced by these two populations sheds light on the current issues of healthcare inaccessibility and declining state of global mental health. COVID-19 containment measures have strained healthcare accessibility, particularly for refugees and displaced persons who already face economic and legal limitations to health services because of loss of livelihood and documentation during displacement (Internal Displacement Monitoring Center [IDMC], 2018). A United Nations report showed that the pandemic affected the work of humanitarian agencies and civil society organizations in conflict settings because of global economic recessions associated with COVID-19; many international donors cut back funding investments for humanitarian programs in low- and middle-income countries (LMICs) (United Nations High Commissioner of Refugees [UNHCR], 2020b). In some cases, general health and political insecurities brought about by COVID-19 hindered many aid agencies and workers from reaching those in need.

In 2020, 85% of the global refugee population were located in LMICs, and the United Nations High Commissioner of Refugees funding gap for forcibly displaced persons was about 50% (UNHCR, 2020a; UNHCR, 2020b). Undoubtedly, displaced persons and refugees who typically live in overcrowded spaces and depend on the informal economy have borne the weight of the crisis. Among these groups, the virus containment restrictions imposed by public health authorities were largely counterproductive to the health and mental well-being of forcibly displaced persons. Consider the efficacy of social distancing measures. While effective in higher-income countries in execution, it is largely impracticable in the close spaces in the typical homesteads and camp settings in which refugees and displaced persons reside in.

### Mental health in COVID-19

WHO has established that mental illness is prevalent in conflict-affected and crisis settings. In 2017, the estimated prevalence rate of mental disorders in conflict-affected low- and middle-income settings was one in five people (Charlson et al., 2019). One recently published review explored mental health issues affecting Africans living in refugee settings during the Ebola crisis and found a high prevalence of depression, anxiety, post-traumatic stress disorder, and obsessive-compulsive disorder among those affected by the disease (Cénat et al., 2020). Since March 2020, analysts have been focused on examining the economic and political implications of

the pandemic. But the mental health dimensions of this crisis, especially for vulnerable persons like those forcibly displaced from their homes, have been understudied. Forcibly displaced persons are more likely to experience unequal access to healthcare, stressful confinements and precarious livelihoods, and are subject to rampant misinformation related to COVID-19. All of these factors can take a toll on mental well-being. Additionally, on account of their precarious status, displaced persons who have been uprooted from their original homelands often lack identity documents and housing security and are thus overlooked or ignored in government-led interventions and public welfare policies. They are prone to a range of human rights violations, forceful containment measures, and race/social-based aggression from host communities. The International Office of Migration of the United Nations (IOM) noted that COVID-19 places psychosocial stressors on displaced persons and argues that governments should invest in evidence-based psychosocial support programs of interventions for forcibly displaced persons (International Office of Migration [IOM], 2021).

### Studying psychosocial well-being for displaced populations in COVID-19

This paper presents a plethora of information on how mental health and well-being have been affected among refugees and internally displaced persons throughout the time of the COVID-19 pandemic, in the midst of aforementioned shifts in national and global dynamics and the limitations of their respective support systems. However, because of the difficulty in comparing the complex national problems and global dynamics of multiple countries, many studies that examine these effects limit their scope to single-country case studies. Out of these single-country case studies, only an occasional few focus on LMICs, with the majority of research studying high-income countries. An encompassing evaluation of refugees and internally displaced persons in global mental health requires a detailed examination of the experiences of these two groups in LMICs. To effectively define the problems of LMICs in approaching these populations, global patterns must be studied to determine common issues, current interventions in the remedy of mental health deterioration must be evaluated, and recommendations for approaching mental health must be explored in the international and national scale. The purpose of this review is to summarize the literature on mental health and psychosocial well-being of refugees and displaced persons who have migrated from LMICs to the same or another LMIC in the context of COVID-19. A systematic review assesses the current literature of the factors behind the psychosocial well-being and mental health of these populations. The themes identified in this process highlight the discouraging situations they face and can influence future approaches to effective care.

### Methodology

A systematic literature review was completed (See Figure 1). A systematic literature review is a comprehensive analysis of the current knowledge on a topic to address an underlying research question (Charles Sturt University, n.d.). For this study, the research question is as follows: *What were the psychosocial- and mental health-related experiences of refugees, asylum seekers, and internally displaced persons who have migrated between LMICs during the pandemic period?*

The inclusion criteria were flexible, permitting consideration of a range of formats, demographic factors and topics that included the

psychosocial and mental health issues of refugees, asylum seekers and/or internally displaced persons. We focused primarily on the time period of the global onset of the COVID-19 pandemic and the implementation of its related restrictions. There were no restrictions on the methodology used by the sources, and we included articles based on quantitative, qualitative and mixed-methods research modalities. We were open to gray literature that may have been found in the databases below (though no gray literature has been used). We only included articles written in English. The populations of interest included refugees, asylum seekers and/or internally displaced persons who have moved from LMICs to other LMICs or within the same LMIC. LMICs in this case are defined by the OECD, or the Organization for Economic Co-operation and Development (OECD, 2020). While there were many sources that described the experiences of these vulnerable populations' migrations concerning countries outside of this list (such as high-middle income countries and high-income countries), these articles were not the main focus of this research and were identified as "Maybe" relevant in article screening for the possibility of comparing results with that of LMICs. However, this comparison proved these articles to be outside the scope of the research, and we rejected them. The topics of interest involved psychosocial and mental health and mental well-being. No specific DSM-5 disorders were used in search terms. Therefore, "psychosocial and mental health" and "well-being" were screened on a general scale to be flexible to culturally different terminologies of mental health.

A search was conducted in five databases: MEDLINE (1946–present via Ovid) (n = 413), EMBASE (n = 587), CINAHL (n = 186), PubMed (n = 906) and PsychINFO (n = 174) with the use of the following search string combinations (formatted for MEDLINE searches, but translated according to the databases above):

1. "Emigrants and Immigrants"/ or Refugees/ or (immigrant* or immigration or emigrant* or emigration or refugee* or "asylum seeker*" or asylee* or "displaced person*" or "displaced people" or "incomer*" or "in comer*" or "new comer*" or newcomer* or migrant* or resettler*

2. Coronavirus/ or exp. Coronavirus Infections/ or (coronavirus* or corona virus* or OC43 or NL63 or 229E or HKU1 or HCoV* or ncov* or covid* or sars-cov* or sarscov* or Sars-coronavirus* or Severe Acute Respiratory Syndrome Coronavirus*).mp.) and (201,906* or 201907* or 201908* or 201909* or 20191* or 2020* or 2021* or 2022* 2023* or 2024* or 2025* or 2026* or 2027* or 2028* or 2029* or 2030*).dt,ez,da.) not (SARS or SARS-CoV or MERS or MERS-CoV or Middle East respiratory syndrome or camel* or dromedar* or equine or coronary or coronal or covidence* or covidien or influenza virus or HIV or bovine or calves or TGEV or feline or porcine or BCoV or PED or PEDV or PDCoV or FIPV or FCoV or SADS-CoV or canine or CCov or zoonotic or avian influenza or H1N1 or H5N1 or H5N6 or IBV or murine corona*).mp.) or (Covid-19/ or covid.mp. or covid19. mp. or 2019-ncov.mp. or ncov19.mp. or ncov-19.mp. or 2019-novel CoV.mp. or sars-cov2.mp. or sars-cov-2.mp. or sarscov2. mp. or sarscov-2.mp. or Sars-coronavirus2.mp. or Sars-

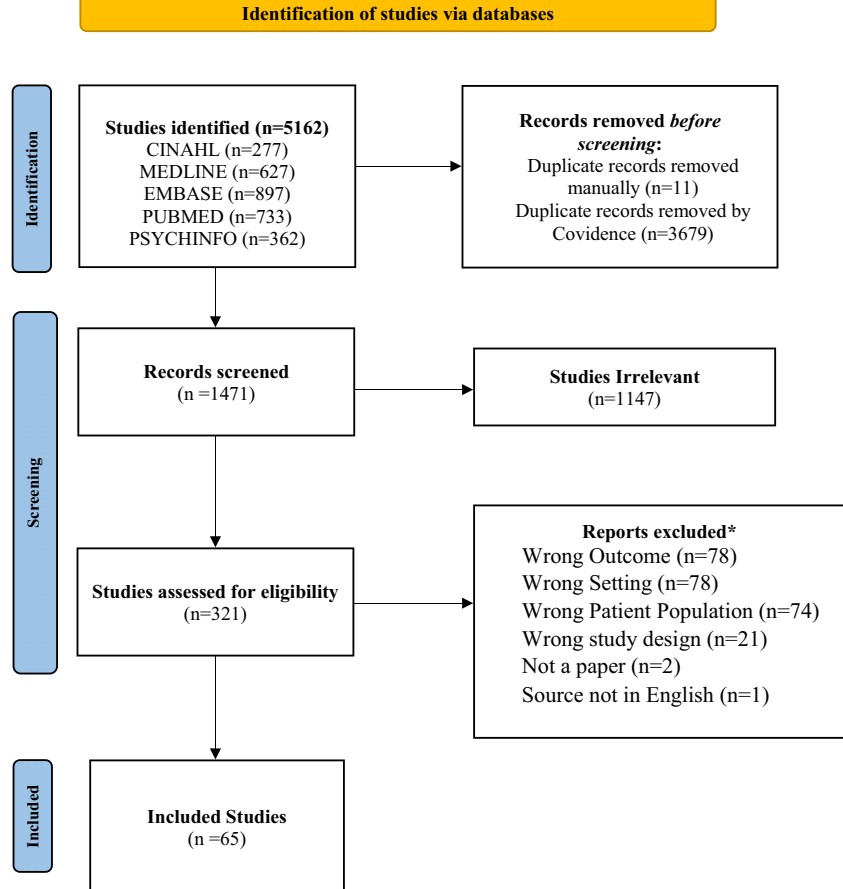

**Figure 1.** PRISMA 2020 flow diagram.

coronavirus-2.mp. or SARS-like coronavirus*.mp. or coronavirus-19.mp. or Deltacron.mp. or Omnicron.mp. or ((novel or new or nouveau) adj2 (CoV or nCoV or covid or coronavirus* or corona virus or Pandemi*2)).mp. or ((subvariant* or variant*) adj2 (India* or "South Africa*" or UK or English or Brazil* or alpha or beta or delta or gamma or kappa or lambda or mu or "AY.X" or "BA.1" or "BA.2" or "BA.3" or "BA.4" or "BA.5" or "P.1" or "C.37")).mp. or ("B.1.1.7" or "B.1.351" or "B.1.617.1" or "B.1.617.2" or "B.1.1.529*" or "B.1.61.7*" or "21 L/BA.2" or "21 K/BA.1").mp.)
3. ("mental* health*" or "Psychosocial wellbeing" or depress* or sadness or stress or anxiety or anxious* or PTSD or "mental* ill*" or "mental* disorder*").

For a full history of the literature search and the specific search strings for each database, please see supplementary material, *Literature Search History.*

A quality appraisal review was conducted using the JBI Critical Appraisal Tools (JBI Global, n.d.); specifically, using their critical appraisal checklists for qualitative research (Lockwood et al., 2015), analytical cross-sectional studies (JBI Global, n.d.), cohort studies (JBI Global, n.d.), prevalence studies (Munn et al., 2015), quasi-experimental studies (Barker et al., 2024), randomized controlled trials (Barker et al., 2023), textual evidence expert opinion (McArthur et al., 2015), and textual evidence narrative (McArthur et al., 2015). Please see *Supplementary Material – Quality Appraisal* for each article's appraisal score. The software Covidence was used for collaborative article review.

## Results

This study synthesizes and summarizes the findings of multiple studies conducted with refugees and displaced persons in LMICs on mental health and psychosocial well-being during COVID-19. Out of the 65 extracted studies, 37 of the studies reported solely on refugees, 6 reported solely on internally displaced persons, 11 of the studies reported solely on vulnerable migrants, and 10 reported on a broad demographic of vulnerable peoples, which includes refugees and internally displaced persons. These studies span multiple countries across multiple geographical areas: Africa, including Burkina Faso, Ethiopia, Ghana, Kenya, Nigeria, Rwanda, Somalia, South Africa, Tanzania, Tunisia and Uganda; Asia, including Bangladesh, China, Indonesia, Malaysia and Myanmar; the Middle East, including Iran, Iraq, Jordan, Lebanon, Palestine and Turkey; and North and South America, including Columbia, Mexico and Peru. Two articles are written in the context of multiple countries; one is written in the context of multiple countries in South America and one within the Turkey–Syrian border. Each of these study characteristics is included in the supplementary material "Extraction Table" (please see Supplementary Material – Extraction Table).

Two main outcomes were found through the review: the first includes the mental and physical health outcomes in relation to the psychosocial challenges faced by the different LMICs' refugee and internally displaced person (IDP) populations (See Figure 2). This includes factors such as COVID regulations, government aids, community, and personal finances, which intersect with one another in different ways across demographics and countries. The second main outcome is the development of interventions in various countries, guided by the needs and psychosocial challenges of their countries, with different focuses and varying degrees of success.

### Theme I: Mental health and psychosocial challenges

These studies highlight the critical influence of various stressors on mental health and well-being of refugees and displaced persons within the COVID-19 context. These stressors intersect according to the context of the diverse studies in this review. While stressors directly related to COVID-19 have a great impact on mental health, a host of other stressors, such as the social structures involving the whole community, lack of government and international aid, as well as the financial capabilities and resource access of these groups, are correlated with adverse outcomes in their health. The physical health and mental health of vulnerable people were damaged by these issues, problems that are exacerbated by the social factors that affect demographics within the international LMIC refugee and IDP populations.

#### Psychosocial stressors within the COVID-19 context
**COVID-19 community measures.** Without a doubt, the physical toll of a COVID-19 infection and the subsequent isolation has a tremendous impact on individual mental health, with an abundance of research dedicated to the effects of individual infection (National Institute of Health, 2023). In a broader scope, COVID-19 has led to adverse outcomes for both refugee and IDP communities, stressors reported in relation to COVID-19 rules and regulations, sickness vulnerability and COVID-related healthcare and information access.

In addition to the highly infectious nature of COVID-19, studies from South Asia and the Middle East have shown the role of living conditions for sickness vulnerability of refugees and IDPs. For example, general living conditions in Palestine facilitated COVID transmission (Jamal et al., 2022). In response to the accelerating spread of COVID-19 infections, countries all over the world implemented public safety measures, including lockdowns within the community, limiting transportation of materials and services, and pressures to vaccinate. One example is Canada's implementation of various legal structures to ensure cooperation with screening officers for travel (Government of Canada, 2020). Despite the intentions of protecting public health, these protocols were additional stressors in an already-hectic, displaced life in countries such as Kenya (Oyekale, 2022), Rwanda (Manirambona et al., 2021), Somalia (Mumin et al., 2022), Bangladesh (Palit et al., 2022), Jordan (Bernardi et al., 2021; El-Khatib et al., 2020; Jones et al., 2022), Columbia (Espinel et al., 2020), Mexico (Piñeiro & Ibarra, 2022) and Iran (Mahmoodi et al., 2023; Khozaie et al., 2024). In some notable cases, the lack of access to COVID-19 information leads to increased distress, such as in Tunisia (Ben Abid et al., 2023) and Turkey (Atak et al., 2023a).

**Social structures involving the community.** Vulnerable populations also reported psychological stressors related to community structures during the COVID-19 crisis period. The pandemic crisis transformed the ways people interacted and influenced the environments of their host countries. Social events with great reach were exacerbated within the refugee and IDP communities in some countries. These include social fallout in Nigeria (Ekoh et al., 2021), national economic hardships in Lebanon (Hajjar & Abu-Sittah, 2021), political stagnation and economic collapse in Lebanon (Fouad et al., 2020), and environments of violence in Mexico (Piñeiro & Ibarra, 2022) and Myanmar (Khai, 2023). A report from Mercy Corps (2021a) reported that the breakdown in social cohesion and increasingly divergent public health knowledge of COVID-19 were associated with conflict-related mental health

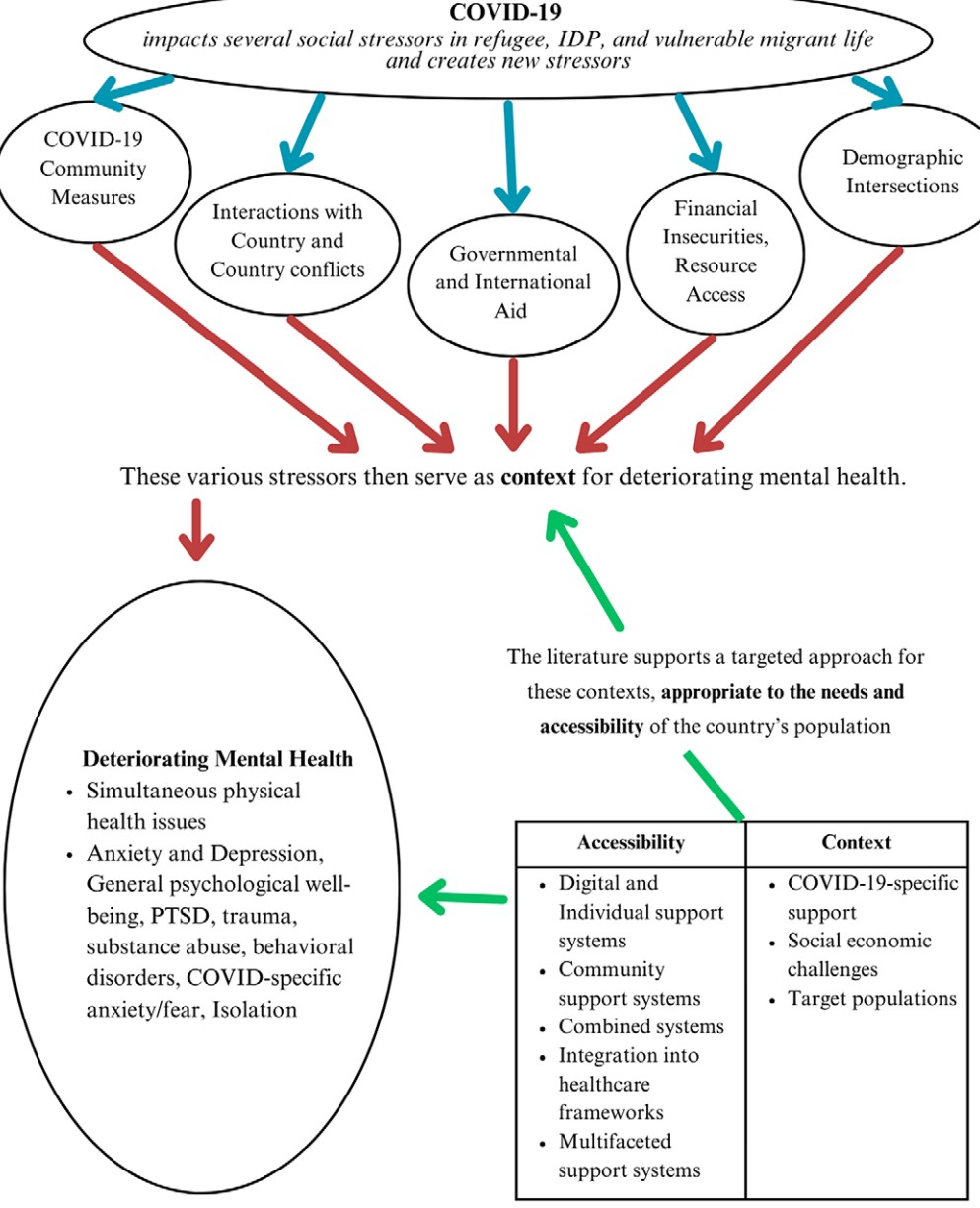

**Figure 2.** Understanding the stressors and interventions of the literature.

problems (Mercy Corps, 2021a). Outside of their settlements waited a world that stigmatized these refugees, who often came from other countries and were believed to be carriers of COVID-19 in countries such as South Africa and Uganda, (Bukuluki et al., 2020; Mukumbang et al., 2020). For example, refugees in Uganda reported stigma and isolation from the host community because they were labeled as potential importers of COVID-19 (Bukuluki et al., 2020). On a global scale, it has been reported that COVID-19 has aggravated ethno-racial tensions and stigmatization of all migrants (Aghajafari et al., 2023) because both the disease and people were perceived by their similarity: their foreign status. Moreover, the unequal distribution of COVID-19 information and testing services was also identified along ethnic and racial lines, demonstrating systemic unevenness in access to health services (Centers for Disease Control and Prevention [CDC], 2021). Building on preexisting social discrimination, structures of stigma

prevailed throughout the pandemic, such as ageism in IDP camps in Nigeria (Ekoh et al., 2021) and Bangladesh (Anwar et al., 2023a), and overall discrimination in Turkey (Kurt et al., 2021) and Iran (Mohammadsadeghi et al., 2022).

*Government programs and international aid.* Prior to the onset of the pandemic, there would be limited access to governmental and international aid programs. However, the regulations caused by COVID-19 further impeded financial assistance access in Burkina Faso (Ozer et al., 2022). It is also not difficult to imagine the hindrance of government processing of refugee and IDP status, as was the case for Uganda (Bukuluki et al., 2020). Specific resources were greatly reduced throughout the pandemic, with reductions in food aid programs and inefficient cash transfer compromises in Rwanda (Manirambona et al., 2021), aid programs being ineffective in improving living conditions in Burkina Faso (Ozer et al., 2022),

and hindrances in identification processing leading to limited food program access in Uganda (Bukuluki et al., 2020). The lack of aid also occurred on a global scale, as humanitarian aid supplies from agencies like the World Food Programme were adversely affected by the economic state of the U.S. and other wealthier donor countries as affected by the pandemic (UNHCR, 2020b; World Food Programme, 2023). The pandemic's threat to aid workers' health security also deterred development work in poorer countries (Manirambona et al., 2021). Despite their importance, health workers' mental well-being in crisis circumstances also calls for consideration as this can impact their capacity to provide care to those most vulnerable. A review of 16 studies on pandemic workplace interventions for health workers dealing with disease outbreaks in past pandemics showed that many settings lack the facilities, time and skills to incorporate mental health programs for health workers needing mental health support (Pollock et al., 2020). As expected, war-torn countries such as Libya were worst hit by COVID-19 restrictions in terms of health systems capacity as conflict and war concurrently occurring with the disease outbreak overburdened health workers. This reportedly led to burnout, verbal abuse and anxiety symptoms among health workers (Elhadi et al., 2020).

Economic capabilities. COVID-19 is associated with financial- and resource-related stressors for refugees and IDPs, further correlated with mental health declines for refugees and IDP populations. Financial insecurities, such as limited money and lower income, were reported for both groups, in countries such as Burkina Faso (Ozer et al., 2022), Nigeria (Ekoh et al., 2021), Uganda (Bukuluki et al., 2020), Indonesia (Hajjar & Abu-Sittah, 2021), Jordan (Akhtar et al., 2021), Turkey (Atak et al., 2023b; İkıışık et al., 2022), Mexico (Piñeiro & Ibarra, 2022), Turkey (Atak et al., 2023b; Budak et al., 2021), and Lebanon (Alnaji et al., 2024; Fouad et al., 2020; Hoffman et al., 2023). Difficulties in employment felt in South Africa (Mukumbang et al., 2020), Somalia (Mumin et al., 2022), Turkey (İkıışık et al., 2022), Jordan (Bellizzi et al., 2021), Columbia (Espinel et al., 2020), and India (Singh, 2021) deepened this insecurity, though a report by Boiko et al. (2024) alludes to the presence of this insecurity in a variety of countries. In a report by Badrfam and Zandifar (2021), these working conditions could also be stressful, as was the case for Iran. These economic hardships could be a factor in the poverty and homelessness faced in South Africa (Mukumbang et al., 2020) and Bangladesh (Hossain et al., 2022). In the suburbs of Kampala city and other towns, the government-imposed lockdown compromised livelihoods and income security for many forcibly displaced communities. The lack of access to food and other basic necessities of life predisposed refugees to malnutrition, anxiety, stress, and psychosocial problems (Hossain et al., 2022).

Resource access. Even after the limitations of resource aid programs, general scarcity of resources has been reported in Burkina Faso (Ozer et al., 2022), Bangladesh (Palit et al., 2022), Jordan (Akhtar et al., 2021), Lebanon (Fouad et al., 2020), India (Ozer et al., 2022), and Turkey (Kurt et al., 2021). This is further specified as food scarcity in Rwanda (Manirambona et al., 2021), South Africa (Mukumbang et al., 2020), Somalia (Mumin et al., 2022), Burkina Faso (Ozer et al., 2022), Venezuela (Espinel et al., 2020), Lebanon (Zeid et al., 2023), and Bangladesh (Hossain et al., 2022; Anwar et al., 2023b). In some cases, post-traumatic stress settled as the shock brought back sour memories of their experiences and ordeals of the life during the conflict in their countries of origin. This finding is consistent with the findings of a previous study by Maharaj et al. (2017), who studied self-reported food insecurity and the risk of depression among refugees and immigrants in South Africa among 335 adults in Durban, South Africa. They found that food insecurity was associated with poverty and mental illness, and micronutrient deficiencies in vitamin B12 and iron were implicated in the etiology of depression among the sampled refugees (Maharaj et al., 2017). General poor living conditions have been reported as a stressor for vulnerable populations in Lebanon (Alnaji et al., 2024), Jordan (Jones et al., 2022), and other countries as alluded to by Boiko et al. (2024). This is also taken further, with overall quality of life being a factor in depressive symptoms for refugees in Turkey (Erol & Batgi, 2023).

Indeed, the weight of these intersecting factors has been highlighted in some literature on refugee and IDP experiences in Bangladesh (Palit et al., 2022) and Iran (Mahmoodi et al., 2023). Financial hardship and resource access are prominently reported for refugees and IDPs in many LMICs in Asia and the Middle East, but especially in African countries such as South Africa, Uganda, and Burkina Faso.

Demographic correlates. Thus far, the aforementioned stressors and adverse health outcomes could be applied with a focus on immigrant status and therefore any person living within the refugee or IDP settlements in their relevant countries. Specific health determinants have been covered throughout the paper thus far, such as immigrant status, resource access and financial aid. Other elements are also correlated with detrimental mental health in these contexts, though to a smaller extent. These included education for vulnerable migrants in Turkey (Atak et al., 2023b; Budak et al., 2021), smoker status in Turkey (Atak et al., 2023a), pregnancy in Turkey (Atak et al., 2023), targeted parental support in North Lebanon (Miller et al., 2022), marital status in Turkey (Budak et al., 2021) and Ghana (Sakyi & Amoako Johnson, 2022), as well as age in Ghana (Sakyi & Amoako Johnson, 2022).

Gender is an interesting health determinant, as it affects refugees, IDPs and migrants in three ways: gender disparities in mental health experiences, gender discrimination and gender violence. Disparities in mental health were found in Uganda (Seruwagi et al., 2022), Bangladesh (Palit et al., 2022), and Lebanon (Hajjar & Abu-Sittah, 2021). Existing literature supports that violence against women is a pervasive practice in contexts of displacement and is associated with poverty and inequitable gender norms (Amodu et al., 2020). Other issues that resulted from economic insecurity during COVID-19 among the displaced included early marriage, transactional sex, dysfunctional family dynamics, and intramarital violence. These themes of gender-based violence were present in Myanmar (Khai, 2023) and Palestine (Jamal et al., 2022). Children were also a vulnerable demographic within refugees and IDPs in LMIC, such as in Jordan (Bellizzi et al., 2021), Tanzania (Shayo et al., 2024), and Lebanon (Hajjar & Abu-Sittah, 2021). In some cases, youths had specific troubles adjusting to their host country during this period, such as limited access to education in Lebanon (Hajjar & Abu-Sittah, 2021) and Tanzania (Shayo et al., 2024) as well as increased experiences of violence in Palestine (Jamal et al., 2022), Tanzania (Shayo et al., 2024), and Myanmar (Khai, 2023).

*Health correlates of psychosocial stressors*
Physical health correlates. Changes brought on by COVID-19 were correlated with adverse physical health outcomes for refugees and IDPs. Physical health also plays a unique role in failing mental health, playing as both a cause and consequence of it. This was

certainly the cause in South Africa, where secondary health concerns were linked to negative coping mechanisms such as substance abuse (Mukumbang et al., 2020) or the simultaneous link of psychological distress and poor physical health, as was the case for refugees in Uganda (Seruwagi et al., 2022). This issue is not limited to physical ailments that were caught during the pandemic's height, such as COVID-19 infections or other sicknesses. Preexisting health issues were reported to be a hindrance in already-fragile mental health, in countries such as Ghana (Sakyi & Amoako Johnson, 2022). The issue of physical health is then worsened by the limited access to healthcare, as experienced by vulnerable groups in South Africa (Mukumbang et al., 2020), Somalia (Mumin et al., 2022), Burkina Faso (Ozer et al., 2022), Lebanon (Hajar & Abu-Sittah, 2021; Alnaji et al., 2024), Iran (Badrfam & Zandifar, 2021), Jordan (El-Khatib et al., 2020), Turkey (İkiışık et al., 2022), and Bangladesh (Hossain et al., 2022; Anwar et al., 2023b).

Mental health and psychological correlates. COVID-19 was reported as the cause of a wide variety of adverse mental health outcomes. Common mental health issues, such as anxiety and depression, were often reported: anxiety was reported in Bangladesh (Hossain et al., 2022), India (Singh, 2021), Syria (Kira et al., 2023), Somalia (Mumin et al., 2022), Kenya (Oyekale, 2022), Ghana (Sakyi & Amoako Johnson, 2022), Turkey (İkiışık et al., 2022; Yalcin et al., 2021), and Uganda (Seruwagi et al., 2022), as well as depression in Turkey (Sevinc et al., 2021), or both anxiety and depression, such as in Turkey (Kurt et al., 2021), or in various countries (Lushchak et al., 2024). However, a few studies did find a little decrease in anxiety and depression rates or no change altogether, such as in Jordan (Akhtar et al., 2021) and Turkey (Atak et al., 2023a; Atak et al., 2023b; Ünver & Perdahlı Fiş, 2022; Yalcin et al., 2021), though these are rare within the literature. Many studies often report general terms for mental health, such as psychological or mental distress, that may not be specific in diagnosis or perceived psychological effects. General mental health decline, stress or psychological distress was reported in Iran (Khozaei et al., 2022; Khozaei et al., 2024), Palestine (Jamal et al., 2022), Rwanda (Manirambona et al., 2021), South Africa (Mukumbang et al., 2020), Bangladesh (Palit et al., 2022) and China (Hall et al., 2021). Other articles also reported intense psychological issues, such as PTSD in Bangladesh (Hossain et al., 2022), Uganda (Baluku et al., 2023a), Iraq (Kizilhan & Noll-Hussong, 2020), Turkey (Seçinti et al., 2023; Yalcin et al., 2021) and various other countries (Lushchak et al., 2024); type III trauma in Syria (Kira et al., 2023); obsessive-compulsive symptoms in various countries (Boiko et al., 2024); and substance abuse disorders and elevated substance use in India (Singh, 2021) and Turkey (Atak et al., 2023a). One notable inclusion is Ünver and Perdahlı Fiş, 2022, as they note the increase of multiple behavioral disorders such as ADHD, ODD and conduct disorder in Turkey. Isolation, found in Uganda (Bukuluki et al., 2020), Nigeria (Ekoh et al., 2021) and Lebanon (Hajjar & Abu-Sittah, 2021; Alnaji et al., 2024), was also reported as an experience resulting from stressors. A few studies approached mental health through the changes in the brain.

Research suggests that the perceived risk of exposure to COVID-19 impacts mental health. For example, Kim et al. (2020) conducted a mixed methods survey to evaluate the mental health impacts of the COVID-19 pandemic in an urban setting in Soweto, South Africa. In the general population surveyed, a higher perceived risk of COVID-19 infection was associated with greater depressive symptoms during the first six weeks of quarantine. In

fact, in examining COVID-19 perceptions, multiple academics report an increase in anxiety directly related to the pandemic and infection in articles from Jordan (Akhtar et al., 2021), Bangladesh (Anwar et al., 2023b), Turkey (Budak et al., 2021), and Tanzania (Shayo et al., 2024).

Certain psychological effects are the result of both COVID-19-related issues and the underlying social structures that refugees and IDPs previously listed. A unique mental health outcome of COVID-19 was the social isolation for refugees in LMICs. Whether this was the result of COVID-19's restrictions or the social exclusion of their host country, the weight of isolation was observed to be influential in the decreasing quality of mental health for these vulnerable groups in Uganda (Bukukluki et al., 2020), Nigeria (Ekoh et al., 2021) and Lebanon (Hajjar & Abu-Sittah, 2021).

Effects of detrimental mental health. The consequences of detrimental mental health can be cyclical in nature and have dire implications within crisis contexts. Two studies, both conducted in Turkey by Alpay et al (2021) and Kira et al. (2021), found that PTSD and depression associated with COVID-19 traumas, stresses and fears have affected deficits in working memory and inhibitory control. In another report from Uganda, Baluku et al. (2023) found that psychological inflexibility correlated with poorer adherence to COVID-19 health measures. This implies that the relationship between COVID-19 and mental health is bidirectional, as a report by Hall et al. (2021) finds that poor confidence in the protective abilities of COVID-19 measures was linked to higher depression.

### Theme II: Interventions to address mental health needs

The revelation of existing psychological challenges and their adverse effects on mental health have led many academics to outline specific issues to address in developing mental health interventions for refugees and IDPs in LMIC. This included broad gaps in existing interventions or the lack thereof as well as brief mentions of individual strategies for maintaining mental health. Other studies take a different approach taking various interventions as a focus, investigating their effects on mental health, livelihoods, and well-being among refugees and internally displaced persons (IDPs). These interventions include digital health initiatives, community-based psychosocial support, and targeted mental health interventions. The studies highlight the critical role of social support and coping strategies in mitigating the pandemic's negative mental health effects.

#### Intervention needs

The prescription of mental health interventions greatly varies according to the needs of the individual vulnerable person. This is further complicated by the different issues each country faces and the different issues between IDPs and refugees. For example, in Indonesia, refugees and IDPs simply did not have enough access to interventions, with issues such as barriers to access consultation in Indonesia (Hoffman et al., 2023) and Jordan (Bellizzi et al., 2021). However, in countries such as Uganda (Bukuluki et al., 2020), Somalia (Mumin et al., 2022), Columbia (Espinel et al., 2020) and Lebanon (Hajjar & Abu-Sittah, 2021), the literature displays the absence of these mental healthcare services altogether. The need for mental health interventions is incredibly important for academics such as Hoffman et al. (2023), and Mumin et al. (2022), who explicitly demand interventions for mental health in Indonesia and Somalia, respectively. Atak et al. (2023a) note the direct

effectiveness of such immediate support services, correlating such with minimal anxiety and depression.

In formulating the ideal mental health intervention, various authors point to various contextual factors that must be taken into consideration. The ideal mental health intervention is a targeted approach, to the context of a given population's socioeconomic determinants and in consideration of accessibility (Baluku et al., 2023b; Budak et al., 2021; Greene et al., 2024; Hall et al., 2021; İkiışık et al., 2022). For example, for IDPs in Myanmar, Lee et al. (2022) state the importance of considering accessibility within a crisis context. This could mean having phone access to mental health interventions, or assigning appropriate frontline workers who can focus on certain mental health issues (Lee et al., 2022). For others, a mental health intervention targets the underlying stressors that these vulnerable groups face rather than their adverse mental health consequences, providing a dual role in treating mental illness and easing their transition into a new country. For example, Ben Abid et al. (2023) and Baluku (2023) point to the COVID-19 context and targeting interventions to aid in COVID information and attitudes. Additionally, social-economic challenges such as in Jordan (Bernardi et al., 2021) and Turkey (Atak et al., 2023b) or, specifically, gender-based differences, physical healthcare and social support (Seruwagi et al., 2022) were also noted as key contexts. For refugees in Iran, this includes access to education, economic opportunities, and social support (Mahmoodi et al., 2023). Immigrant status and the transitory period are also reported as needing to be considered, with more immediate interventions for language barriers (Atake et al., 2023a) or general adaptation (Boiko et al., 2024), as a precursor to long-term social-economic challenges. Some studies reveal that their target receivers depended on the specific context of the countries' populations. For instance, the interventions targeted Syrian refugees in Lebanon, while the focus was on internally displaced persons in Myanmar (Cuijpers et al., 2022, Lee et al., 2022). These theoretical considerations are key to the creation of mental health interventions that are practical in effect and in cost.

## Discussion

### Overall patterns

For many countries, the studies reviewed illustrated the need for a host of resources, including food, water supplies, sanitary supplies, mobile clinics, and solid waste disposal. However, some studies in the literature indicate a gap in the provision of these resources on both the global and national scale. The literature also demonstrates the profound impact COVID-19 continues to have on mental and psychosocial well-being of refugees and internally displaced persons. The concepts of psychosocial well-being and mental health are sometimes used interchangeably; however, in the context of a crisis like COVID-19, it is important for public health experts to differentiate between the immediate psychosocial shock of COVID-19 and the global mental health outcomes associable to pandemic-related restrictions. It is imperative that studies explore the long-term impact of the pandemic restrictions on people's mental health, especially how social demographics are affected as a focus.

### Analysis of mental health interventions

In response to the consequences of COVID-19, LMICs have implemented various restrictions in mitigating the outbreak of the pandemic. Mental health interventions are necessary, as there is a rise in negative coping strategies such as substance, abuse, lack of self-care

and isolation and a decrease in community-based activities that provide a sense of support due to COVID-19 social gathering restrictions. Many factors are important in considering the various contexts of mental health treatment, including the type of vulnerable population (refugees or internally displaced persons) as well as the various priorities of the different LMICs. This consideration of priorities examines whether mental health treatment is a relatively high priority of the country in comparison to other initiatives, such as food and housing provision for these populations. An intersectional policy approach to addressing the social and ecological determinants of mental health outcomes should address these complex factors that affect mental health, including consideration for gender, housing, employment status, gendered predisposition to social isolation, social belonging needs and social inequities as interrelated influences on mental health (Kola et al., 2021). In terms of global mental health funding, although 116 (89%) countries reported that mental health and psychological support was part of their national COVID-19 response plans, only 17% said they had committed additional funding for this. A WHO survey of 28 African countries on regional access to mental health services showed that 37% of African states with a mental health response plan had only partial funding and an additional 37% reported having no funds at all (WHO, 2020b). This review highlights the need to address economic constraints to overall psychological well-being of refugee populations.

### Possible interventions

There are a few methods in reaching the right populations for support. These interventions, which included digital health initiatives, community-based psychosocial supports and targeted mental health interventions, highlight the importance of innovative and adaptable approaches to ensuring accessibility and effectiveness in a variety of settings. Digital interventions were seen to be effective in Peru (Greene et al., 2024) and Lebanon (Cuijpers et al., 2022). This creative approach demonstrates the potential of digital solutions to bridge gaps in mental health service delivery, especially during times of crisis when traditional methods may be inaccessible or unsafe.

Many articles discuss the effectiveness of community support specifically. General community support has been a great buffer against adverse mental health outcomes, as seen in Syria (Kira et al., 2023), Tunisia (Ben Abid et al., 2023), Turkey (Kurt et al., 2021), and Ethiopia (Desie et al., 2021). The community-based psychosocial support focal point response in Myanmar demonstrates the importance of utilizing local resources and expertise to provide mental health support to IDPs (Lee et al., 2022). The intervention established a resilient, sustainable and culturally sensitive and relevant domestic support system for IDPs in need. In contrast, the lack of social support has intensified the deterioration of mental health from societal pressures, as seen in Uganda and Indonesia (Bukuluki et al., 2020; Hoffman et al., 2023). Certain case studies of initiatives within some LMICs pinpoint these interventions' benefits and successes, as well as present areas for further development. The context-dependent application of services can be seen by comparing the interventions of Lebanon and Myanmar, which required innovative and adaptive approaches to ensure their effectiveness and accessibility (Cuijpers et al., 2022, Lee et al., 2022).

Though optimistic, not all vulnerable groups can find this same success. In a study of vulnerable groups in South Africa, Mukumbang et al. (2020) found that negative coping mechanisms such as substance use have led to further detriment of mental health. An alternate solution could be further displacement to escape social

problems and adverse mental health. This was the case for IDPs in Nigeria and Rwanda, who were further displaced or returned to their origins due to stressors and insufficiently met needs (Ekoh et al., 2021; Manirambona et al., 2021). In many informal interventions, the onus of mental health management becomes another task in an already long list of duties for refugees and IDPs, with no guarantee of success. It is imperative that international or national organizations of refugee aid focus on developing interventions that address mental health struggles for refugees and IDPs as a whole.

The combined efforts between community support and digital interventions lend credibility to the multifaceted approaches to interventions that some authors, such as Jones et al. (2022) and Atak et al. (2023b), allude to. Not only is this effective on a practical level, as seen in various countries' responses to the UNHCR support (Çalıyurt, 2021; Tay & Balasundaram, 2021), support measures in Jordan (El-Khatib et al., 2020) and supports in various South American countries (Greene et al., 2024).

## Conclusions

There is limited peer-reviewed research on psychosocial well-being and mental health of forcibly displaced persons in the context of COVID-19, especially in LMICs. The pandemic's legacy has challenged the capacity of research, healthcare institutions and global powers to identify, understand and address health service gaps. Addressing the psychological stressors of refugees and IDP populations requires the consideration of contextual and innovative approaches to care.

This overarching pandemic mental health problem should prompt nations to set upstream targets to improve living conditions and income security and improve investments in addressing sociocultural determinants of mental health for these vulnerable populations.

**Open peer review.** To view the open peer review materials for this article, please visit http://doi.org/10.1017/gmh.2024.110.

**Supplementary material.** The supplementary material for this article can be found at http://doi.org/10.1017/gmh.2024.110.

**Data availability statement.** Data availability is not applicable to this review article as no new data were created or analyzed in this study.

**Acknowledgements.** We would like to thank Jackie Stapleton of the University of Waterloo, in the Information Services and Resources division of the Davis Centre Library, as well as Megan Kennedy of the University of Alberta, in the Health Sciences Division of J. W. Scott Library, for their guidance in the research process of this project. We would also like to thank Bernard Mensah, whose work was influential throughout the development of the paper. We also thank N. Armoush and Y. Kou for their efforts in updating the review to July 10th, extracting the new papers and completing the quality appraisal.

**Author contribution statement.** O.A. and C.J. conceptualized the original scope of the research, through the study of the same populations in African countries, and the expansion of global refugees and internally displaced persons and presented this idea to K.T.P., acquiring funding and resources to begin the project. O.A. conducted the research and wrote the first draft of the study in African contexts.
Throughout the development of the paper, O.A. has maintained project administration through the management of the research's execution, providing supervision to the research team and advice where necessary, and alongside Bernard Mensah extracted information from the data search in the production of the final study's results, and all authors drafted the final study and revised the article. O.A. and K.T.P. developed the search strategy and data search, with all authors

contributing to the refinement of the data. K.T.P. facilitated the processing of the final article as the corresponding author.

**Financial support.** We express our sincere gratitude for the financial support from the University of Waterloo, through their AMTD Postdoctoral Research Funds. Their continued commitment to advancing knowledge in mental health and global refugees and internally displaced persons' well-being is greatly appreciated.
We have no known conflict of interest to disclose.

**Competing interest statement.** The authors certify that they have no conflicts of interest concerning the research, authorship and publication of this manuscript. All authors have contributed significantly to the development of the study and have approved the final version of the manuscript for submission. The content of this paper was not influenced or biased by any financial, interpersonal or professional affiliations.

**Ethics statement.** This article is the synthesis of previously established results of other journal sources; as such, it does not involve direct research with human participants, tissue or data, clinical health-related research or animal research, and any ethics approval in these domains is irrelevant to the study.

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
