## [Editor Report]

Thanks to the authors for submitting this narrative review. As noted by the reviewers, this is an important topic and having a review synthesizing the literature on mental health among IDPs and refugees during the COVID-19 pandemic is a valuable contribution to the literature. However, both reviewers have suggested some recommendations for restructuring aspects of the review, justifying and describing the methodological approach, as well as how the results are interpreted in the discussion/conclusions. I encourage the authors to consider revising their article according to these suggestions.

---

## [Editor Report]

Thank you for resubmitting your manuscript to Global Mental Health. As noted by the reviewer, this revised draft addresses many of their major concerns. However, two major concerns still remain, which I hope the authors will consider: 1) incorporating a quality appraisal of included studies; and 2) updating the review. Please also check the inconsistencies the reviewer has highlighted in the supplementary information.

---

## [Editor Report]

Thank you for thoroughly responding to the suggestions provided by the reviewer, particularly the addition of the quality appraisal. There are no remaining suggestions or concerns. Thank you for submitting your manuscript to Global Mental Health.